# Dual Molecular Diagnoses of Recessive Disorders in a Child from Consanguineous Parents: Case Report and Literature Review

**DOI:** 10.3390/genes13122377

**Published:** 2022-12-16

**Authors:** Gabriela Roldão Correia-Costa, Ana Mondadori dos Santos, Nicole de Leeuw, Sumara Zuanazi Pinto Rigatto, Vera Maria Santoro Belangero, Carlos Eduardo Steiner, Vera Lúcia Gil-da-Silva-Lopes, Társis Paiva Vieira

**Affiliations:** 1Department of Translational Medicine—Medical Genetics and Genomic Medicine, School of Medical Sciences, State University of Campinas, Campinas 13083-887, São Paulo, Brazil; 2Department of Human Genetics, Radboud University Medical Center, P.O. Box 9101, 6500 HB Nijmegen, The Netherlands; 3Department of Pediatrics, School of Medical Sciences, State University of Campinas, Campinas 13083-887, São Paulo, Brazil

**Keywords:** dual molecular diagnoses, recessive inheritance, consanguineous parents, primary microcephaly, nephropathic cystinosis, *ASPM*, *CTNS*

## Abstract

The widespread use of whole exome sequencing (WES) resulted in the discovery of multilocus pathogenic variations (MPV), defined as two or more distinct or overlapping Mendelian disorders occurring in a patient, leading to a blended phenotype. In this study, we report on a child with autosomal recessive primary microcephaly-5 (MCPH5) and nephropathic cystinosis. The proband is the first child of consanguineous parents, presenting a complex phenotype including neurodevelopmental delay, microcephaly, growth restriction, significant delay of bone maturation, lissencephaly, and abnormality of neuronal migration, photophobia, and renal tubular acidosis. WES revealed two pathogenic and homozygous variants: a c.4174C>T variant in the *ASPM* gene and a c.382C>T variant in the *CTNS* gene, explaining the complex phenotype. The literature review showed that most of the patients harboring two variants in recessive disease genes are born to consanguineous parents. To the best of our knowledge, the patient herein described is the first one harboring pathogenic variants in both the *ASPM* and *CTNS* genes. These findings highlight the importance of searching for MPV in patients with complex phenotypes investigated by genome-wide testing methods, especially for those patients born to consanguineous parents.

## 1. Introduction

Medical genetics has experienced an unprecedented transformation since the advent of genome-wide, next-generation sequencing techniques. Even for the most complex phenotypes, the accelerated discovery of novel candidate disease genes has unraveled disease mechanisms and phenotypes. The widespread use of whole exome sequencing (WES) in both diagnostic and research assays has revealed new pathogenic variants for many conditions, and uncovered unique interplays between these variants in different genes in parallel, enabling a genotype–phenotype correlation in many heterogeneous and complex disorders [1,2].

More than identifying novel disease genes and helping elucidate distinct genetic mechanisms within single genes, WES has created a paradigm shift in rare disease research and diagnosis by providing insights into the mechanisms that interlink dual or even multiple genomic variations and diseases [1,2,3]. The so-called multilocus pathogenic variations (MPV) are a form of mutational burden and represent two or more distinct or overlapping Mendelian disorders that occur either concurrently or develop sequentially over time, leading to blended phenotypes [2].

MPV can lead to distinct or overlapping expression of Mendelian disease traits and their inheritance pattern depends on the substructure of the studied population. A recent review regarding this theme showed that homozygous variants in genes with recessive Mendelian inheritance were more common in children born to consanguineous parents [2,4]. Genome and exome sequencing have identified MPV in 1.4% to 22% of patients with rare diseases [1,5,6,7,8,9,10,11,12]. In nonconsanguineous populations, the simultaneous finding of two or more diseases with autosomal dominant (AD) inheritance is almost five times more frequent than finding diseases with autosomal recessive (AR) inheritance [1]. On the other hand, in consanguineous populations, the prevalence of MPV cases with homozygous variants in AR disease genes can reach up to 29% of patients [12].

In this study, we report on a child, born to consanguineous parents, with dual molecular diagnoses: autosomal recessive primary microcephaly-5 (MCPH5) and nephropathic cystinosis, due to homozygous pathogenic variants in the *ASPM* and *CTNS* genes, respectively. We also performed a literature review regarding cases of MPV in disease genes with recessive inheritance.

## 2. Materials and Methods

### 2.1. Chromosomal Microarray Analysis (CMA)

CMA was performed as a first-tier test using the CytoScan HD Array from Affymetrix^®^ (Thermo Fisher Scientific Inc.—Life Technologies, Carlsbad, CA, USA) following the manufacturer’s instructions. The data were analyzed using the Affymetrix^®^ Chromosome Analysis Suite (ChAS) version 4.0 (Thermo Fisher Scientific Inc.—Life Technologies, Carlsbad, CA, USA). The interpretation and classification of copy number variants (CNVs) and regions of homozygosity (ROH) were performed as previously described [13,14] following recommendations from the American College of Medical Genetics and Genomics(ACMG) [15] and the European guidelines for constitutional cytogenomic analysis [16]. The percentage of homozygosity in the patient’s autosomal genome was calculated by the sum of all homozygous regions detected (excluding the sex chromosomes), divided by total autosomal length, and multiplied by 100 [17] after using fixed detection settings.

### 2.2. Whole Exome Sequencing (WES)

WES was carried out in the Genomic Diagnostics Division of the Human Genetics Department at the Radboud University Medical Center (RUMC) in Nijmegen, the Netherlands. Exome capture was performed using the Agilent SureSelect Target Enrichment V5 (Agilent Technologies, Santa Clara, CA, USA) capture kit, followed by sequencing on the Illumina HiSeq platform (Illumina, San Diego, CA, USA) with 101bp paired-end reads to a median coverage of 75x. The BWA Aligner (version 0.5.9-r16) was used to align the sequence reads to the hg19 reference genome, and variants were called by the GATK unified genotyper (version 3.2-2). The annotation was performed using the laboratory’s custom diagnostic annotation pipeline [18].

The interpretation of variants from WES was performed as previously described by Lelieveld et al. (2016) [18], first using a panel of genes for intellectual disability (Genome Diagnostics Nijmegen-Gene Panel: Intellectual Disability; version DG 2.16) and, secondly, a panel for all OMIM genes (the Mendeliome gene panel; version DG 2.16). Nucleotide variant classification was completed according to the ACMG recommendations [19]. Similarly, the nomenclature of each sequence variant described in this study followed the Human Genome Variation Society—HGVS guidelines.

### 2.3. Sanger Sequencing

The confirmation of WES findings and segregation analysis was performed by Sanger sequencing, according to standard procedures, using the automatic capillary electrophoresis sequencer ABI 3500 Genetic Analyzer (Thermo Fisher Scientific Inc.—Life Technologies, Carlsbad, CA, USA) and the BigDye™ Terminator v3.1 Cycle Sequencing Kit (Thermo Fisher Scientific Inc.—Life Technologies, Carlsbad, CA, USA). The CodonCode Aligner (version 8.0.2) program was used to analyze data.

### 2.4. Literature Review

For the literature review, we searched for articles published on PubMed up to October 2022. The terms of the search strategy used were “dual molecular diagnoses”, “multiple pathogenic variants”, and “blended phenotypes”. We also searched for similar articles in PubMed and the reference list of the selected papers. Only cases with dual or multiple molecular diagnoses of recessive disease genes were included.

## 3. Results

### 3.1. Case Presentation

The proband is a girl, referred for genetic evaluation at two years of age, due to neurodevelopmental delay and microcephaly. She is the first child of healthy consanguineous parents (first cousins). Her mother was also born to consanguineous parents (Figure 1D). During pregnancy, maternal cytomegalovirus infection was diagnosed. She was born by cesarean section due to a significant intrauterine growth restriction and oligohydramnios at 36 weeks of gestation. Birth measurements were 2095 g weight (−1.85 SD), 42 cm length (−2.78 SD), and head circumference of 27 cm (−4.02 SD); APGAR scores were 8 and 9 at 1 and 5 min, respectively. No neonatal intercurrence was observed and serology for cytomegalovirus and tomography scan of the skull did not show evidence of congenital infection.

During the first month of life, she presented with severe hypotonia, neurodevelopmental delay, and important growth restriction. At 2 years and 4 months of age, her weight was 6000 g (−6.5 SD), length 72.5 cm (−4.9 SD), and head circumference 34.2 cm (−9.6 SD). Physical examination showed no relevant dysmorphisms (Figure 1A). Complementary exams included echocardiography, dilated-pupil fundus examination, and ultrasound of the urinary system, which demonstrated unchanged pathways at that time. Subsequently, radiography showed bowing of the long bones in the upper and lower extremities, demonstrating Madelung deformity, and an important delay of bone maturation associated with abnormal lower limb bone morphology. Brain nuclear magnetic resonance showed parenchyma with volumetric decrease, lobes of the telencephalon with smooth appearance (lissencephaly), abnormality of neuronal migration, and the presence of shallow grooves (Figure 1E).

At 6 years of age (Figure 1B), the patient has developed photophobia and stage 4 chronic renal failure. Fanconi Syndrome (glucosuria, aminoaciduria, metabolic acidosis) was detected and she presented with hypothyroidism, which strongly suggested the diagnosis of Nephropathic Cystinosis. An ophthalmological examination with a slit lamp revealed cystine crystals in the cornea, confirming the diagnosis. At this age she started receiving oral cysteamine. Because of severe glomerular damage, she started undergoing peritoneal dialysis. At the last clinical evaluation, at 9 years of age, photophobia and palpebral edema were evident (Figure 1C). The chronic renal failure worsened, and successful kidney transplantation was performed at 10 years of age. The use of oral cysteamine was maintained to control cystine accumulation in other tissues. She is also receiving cysteamine eye drops to relieve photophobia.

### 3.2. Genetic Tests

#### 3.2.1. CMA

The CMA did not show pathogenic or probably pathogenic CNVs. As expected, it revealed ROH throughout the genome that totaled 168.42 Mb in length, representing 5.57% of the autosomal genome, which is within the proportion of expected ROH for children born to first-cousin couples [14].

#### 3.2.2. WES

Whole exome sequencing identified pathogenic single nucleotide variants (SNVs), both nonsense and homozygous, in two genes: *ASPM:NM_018136.5:exon18:c.4174C>T:p.(Arg1392Ter)*, and *CTNS:NM_004937.3:exon7:c.382C>T:p.(Gln128Ter).* The c.4174C>T variant in the *ASPM* gene (ACMG criteria PVS1, PM2 and PM3) is located in the 18th exon of this gene and has not been described in the literature before. This variant is present in a heterozygous state in seven individuals in the Genome Aggregation Database—GnomAD [20]. The c.382C>T variant in the *CTNS* gene (ACMG criteria PVS1, PM2, PM3, and PP5) is located in the 7th exon of this gene and has previously been reported by Town et al. (1998) [21] in association with nephropathic cystinosis and is described as pathogenic in two additional patients according to the ClinVar database. This variant is absent in population genomic databases such as gnomAD [20], 1000Genomes [22], or ABraOM [23].

In addition, a homozygous missense variant in exon 9 of the *DHCR7* gene was found (*DHCR7:NM_001360.3:exon9:c.988G>A:p.(Val330Met)*), which was initially classified as a variant of uncertain significance (VUS-ACMG criteria PM1, PM2, PP3 and BS2) and has been previously reported in association with Smith–Lemli–Opitz (SLO) syndrome [24]. This is considered a rare variant, presenting a frequency of 0.07% in gnomAD [20] and 0.08% in ABraOM [23]. All these genes are encompassed within the ROH detected in the CMA.

#### 3.2.3. Sanger Sequencing

Sanger sequencing confirmed all of the variants in a homozygous state and revealed that both parents are carriers (heterozygous) of the *ASPM* and *CTNS* variants, following Mendelian expectations for an autosomal recessive (AR) trait (Figure 2). However, the *DHCR7* variant was found in heterozygous state in the patient’s father and homozygous state in her mother, who is also born to consanguineous parents and presents a normal phenotype. In addition, biochemical analysis revealed levels of 7-dehydrocholesterol (7-DHC) equal to 1.6 mg/dL (laboratory reference value ≤1.5 mg/dL), which was considered a no significant change and not compatible with SLO diagnosis. Furthermore, clinical reevaluation did not show specific clinical signs of SLO Syndrome.

### 3.3. Literature Review

From the literature review we found 20 studies describing patients harboring biallelic variants in two or multiple disease genes of recessive inheritance, representing 106 individuals (Appendix A). Among these cases, 84 presented dual recessive diagnoses (AR + AR) and 22 presented multiple molecular diagnoses, with at least two recessive diagnoses (AR + AR + _). Furthermore, 72.64% (77 of 106) of these patients were known to be children of consanguineous parents [1,2,3,4,11,12,25,26,27,28,29,30,31,32,33,34,35,36,37]. None of the reported cases carried recessive pathogenic variants simultaneously in the *ASPM* and *CTNS* genes, as the patient herein described.

## 4. Discussion

The application of genome-wide screening methods, such as exome or genome sequencing, has uncovered two or more concomitant pathogenic variants on distinct loci in approximately 5% of the patients suspected of a genetic disorder [1]. Although most of the patients harboring MPV present monoallelic variants in genes of dominant inheritance, some of them present two biallelic variants in disease genes of recessive inheritance. In the last case, parental consanguinity has been found in at least half of these families [38]. The higher risk of diseases with recessive inheritance is well known for children born to consanguineous parents, mainly due to the sharing of multiple common ancestor alleles in the couples’ DNA that will be inherited as homozygous pairs by the progeny [38,39]. Furthermore, consanguinity appears to increase the chance of MPV occurring in a family [1,3].

In this study, we describe a patient presenting a complex phenotype, born to consanguineous parents (first cousins), harboring homozygous pathogenic variants in two recessive disease genes (*ASPM* and *CTNS*). She presented with clinical features compatible with defects in both genes, which are Microcephaly 5–primary-autosomal recessive (MCPH5–OMIM #608716) and Cystinosis-nephropathic (OMIM #219800).

The *ASPM* gene is the human ortholog of the Drosophila melanogaster “abnormal spindle” gene. Its main function is to regulate the mitotic spindle and coordinate the mitotic processes in embryonic neuroblasts, being essential for central nervous system development [40]. Deleterious variants in the *ASPM* gene are the cause of autosomal recessive primary microcephaly (MCPH), which is a neurodevelopmental disorder that exhibits genetic heterogeneity, being associated with variants in at least 20 recessive loci [40,41,42,43,44,45,46]. Among all variants reported so far, those in the MCPH5 locus, which is located in the band q31.3 of chromosome 1, involving the *ASPM* gene, were demonstrated to be the most prevalent cause of MCPH in consanguineous families [47].

The *CTNS* gene is located in the band p13.2 of chromosome 17 and its main function is to encode a membrane protein responsible for cystine transport. There are at least 166 disease-causing variants in this gene reported in individuals with nephropathic cystinosis in the Human Gene Mutation Database (HGMD^®^ Professional 2022.3). In addition, variants in this gene represent the most common inherited cause of renal Fanconi syndrome in children [48,49]. As reported by Town et al. (1998) [21], the *CTNS* variant detected in our patient gives rise to a recessive disorder called cystinosis that results from a defective lysosomal transport of cystine. The predominant pathological finding is the presence of cystine crystals in almost all cells and tissues, especially in the cornea and kidney, where crystal accumulation increases with age [50].

The variants identified in both *ASPM* and *CTNS* genes in the present case are nonsense (truncating mutations), which by itself strongly supports their deleterious effects. Besides that, by carefully analyzing the phenotypic characteristics of the patient, in comparison with clinical findings described in other patients from the literature also harboring pathogenic variants in the *ASPM* and *CTNS* genes, it becomes evident that both variants are contributing to the phenotype of the proband (Table 1). On the other hand, clinical reevaluation, biochemical analysis of 7-DHC, and finding the same homozygous variant in the *DHCR7* gene in her healthy mother do not support that this variant is contributing to the patient’s phenotype. These findings support that the c.988G>A variant in exon 9 of the *DHCR7* gene is not deleterious and is in line with more recent findings by Saskin et al. (2017) [51], Kars et al. (2021) [52], and Aguiar et al. (2022) [53], in which this variant was classified as class 3.

The identification of the cause of the disorders in this patient allowed proper genetic counseling for the family. Although the recurrence risk for each condition is 25%, since the couple are carriers of pathogenic variants in at least two genes, they present a higher risk of having another affected child with one of these disorders. The knowledge of the molecular diagnosis in this family can help in preventing these diseases in future children. This can be accomplished by prenatal genetic testing or, more appropriately, by preimplantation genetic testing and in vitro fertilization. Even though these are recessive disorders, other relatives can be tested for these specific variants in the *ASPM* and *CTNS* genes, for preventing the birth of other affected children.

A retrospective analysis executed by Balci et al. (2017) [3], using data from clinical whole-exome sequencing, indicated that among 802 probands, eight patients (3.5% of 226 diagnosed, 1.0% of total) presented more than one pathogenic variant in different disease-associated genes, each one explaining at least part of their clinical presentation. In seven of the eight families described in this retrospective study, the presumed mode of inheritance was autosomal recessive (AR) for at least one of the variants. Four of these eight families were known to be consanguineous. Posey et al. (2017) [1] also conducted a retrospective analysis of data from whole-exome sequencing of 7374 unrelated patients and showed that among 2076 (28.2%) cases with a molecular diagnosis, 101 (4.9%) had pathogenic variants in two or more disease loci. Two (dual) molecular diagnoses were reported in 97 of these 101 patients, 9 of which were AR + AR.

Studies involving rare phenotypes investigated by WES confidently established that, due to the presence of increased regions of homozygosity in their genomes, children of consanguineous couples are more susceptible to the presence of homozygous pathogenic variants in the genome, either as a single genetic locus variant or as MPV [1,10,33,56]. The detection of multiple relevant findings in a patient’s DNA, besides being rare, requires meticulous teamwork involving the diagnostic laboratory and the clinic, to evaluate the influence of each variant in the phenotype, as well as to define whether one variant fits more or if the patient has multiple conditions [4]. In this context, the availability of WES is a great ally, especially by its ability to simultaneously detect multiple pathogenic variants in blended phenotypes in a hypothesis-free manner [10].

## 5. Conclusions

To the best of our knowledge, the patient herein described is the first one harboring pathogenic variants in both the *ASPM* and *CTNS* genes. The literature review showed that most of the patients harboring two variants in genes of autosomal recessive inheritance are born to consanguineous parents. Our data highlight the importance of searching for MPV in patients with complex phenotypes investigated by genome-wide testing methods, especially for those patients born to consanguineous parents.

## Figures and Tables

**Figure 1 genes-13-02377-f001:**
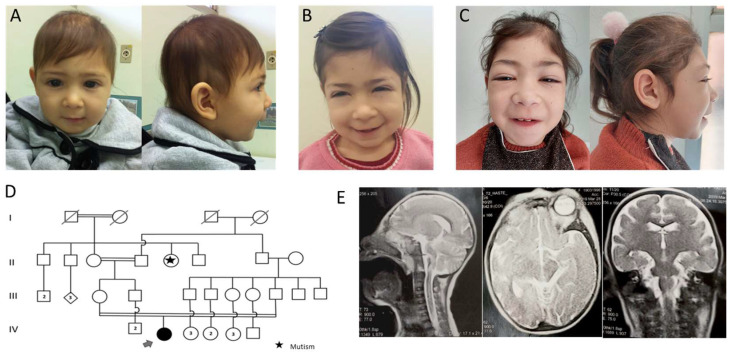
(**A**–**C**) Pictures of the patient at 2 years, 6 years, and 9 years of age, respectively, showing no relevant dysmorphisms besides microcephaly. Closed eyes point to photophobia. (**D**) Four-generation pedigree of family showing multiple consanguineous unions. Proband’s parents are first cousins and her mother was also born to consanguineous parents. (**E**) Brain nuclear magnetic resonance showing parenchyma with volumetric decrease and lissencephaly.

**Figure 2 genes-13-02377-f002:**
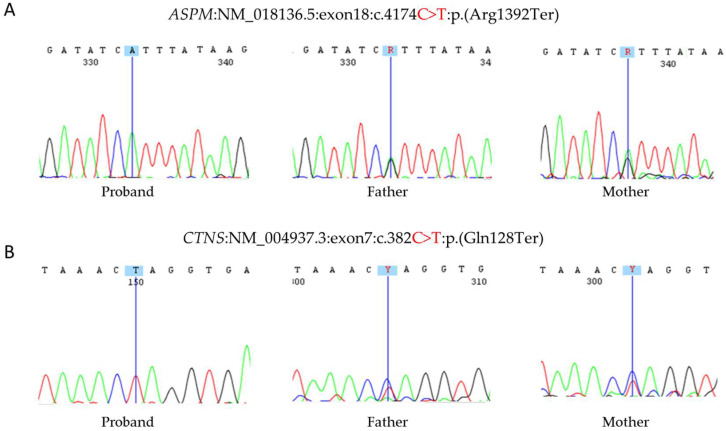
Sequence electropherograms of the trio for both variants. (**A**) c.4174C>T in the *ASPM* gene in homozygous state in the patient and heterozygous state in her parents. (**B**) c.382C>T in the *CTNS* gene in homozygous state in the patient and heterozygous state in her parents.

**Table 1 genes-13-02377-t001:** Genotype–phenotype correlation for *ASPM* and *CTN* genes.

Patient’s Clinical Findings	*ASPM* Phenotype *	*CTNS* Phenotype *
Growth		
Birth length less than 3rd percentile (HP:0003561)	+	+
Failure to thrive in infancy (HP:0001531)	na	+
Neuromotor development and neurological features		
Motor delay (HP:0001270)	+	na
Delayed speech and language development (HP:0000750)	+	na
Intellectual disability, severe (HP:0010864)	+	na
Hyperactivity (HP:0000752)	-	na
Attention deficit hyperactivity disorder (HP:0007018)	-	na
Myopathy (HP:0003198)	na	-
Seizure (HP:0001250)	-	na
Progressive neurologic deterioration (HP:0002344)	na	+
Central Nervous System abnormalities		
Aplasia/Hypoplasia of the corpus callosum (HP:0007370)	+	na
Small cerebral cortex (HP:0002472)	+	na
Lissencephaly (HP:0001339)	+	na
Hypoplasia of the pons (HP:0012110)	-	na
Hypoplasia of the frontal lobes (HP:0007333)	+	na
Aplasia/Hypoplasia of the cerebellum (HP:00007360)	+	na
Ventriculomegaly (HP:0002119)	-	na
Cerebral atrophy (HP:0002059)	na	+
Head and neck		
Primary microcephaly (HP:0011451)	+	na
Sloping forehead (HP:0000340)	+	+
Narrow forehead (HP:0000341)	-	na
Proptosis (HP:0000520)	-	na
Highly arched eyebrows (HP:0002553)	-	na
Ears		
Hearing impairment (HP:0000365)	-	na
Eyes		
Photophobia (HP:0000613)	na	+
Peripheral retinal degeneration (HP:0007769)	na	+
Visual loss (HP:0000572)	na	+
Corneal crystals (HP:0000531)	na	+
Voice		
Weak voice (HP:0001621)	na	-
Skeletal		
Delayed skeletal maturation (HP:0002750)	na	+
Genu valgum (HP:0002857)	na	+
Digestive system		
Hepatomegaly (HP:0002240)	na	+
Urinary system		
Renal Fanconi syndrome (HP:0001994)	na	+
Stage 5 chronic kidney disease (HP:0003774)	na	+
Nephrolithiasis (HP:00000787)	na	nr
Hypophosphatemic rickets (HP:0004912)	na	+
Episodic metabolic acidosis (HP:0004911)	na	+
Polyuria (HP:0000103)	na	+
Generalized aminoaciduria (HP:0002909)	na	+
Renal Hypophosphatemia (HP:0008732)	na	nr
Hyponatremia (HP:00029202)	na	+
Microscopic hematuria (HP:0002907)	na	+
Endocrine System		
Primary hypothyroidism (HP:0000832)	na	+
Exocrine pancreatic insufficiency (HP:0001738)	na	+

na = not applicable/nr = not reported. * According to The Human Phenotype Ontology (HPO) catalog [54] and GeneReviews^®^ [55].

## Data Availability

The data that support the findings of this study are available from the corresponding author upon reasonable request.

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
