# Peer review of "Dual Molecular Diagnoses of Recessive Disorders in a Child from Consanguineous Parents: Case Report and Literature Review"

_genes, 2022, doi:10.3390/genes13122377_

Round 1

Reviewer 1 Report

Figure 2: I suggest the authors to improve the quality of Figure 2 by standardizing the text contained in it and explaining in the legend the parts of the sequences highlighted in green (Figure 2B); Page 8, two lines before the discussion, a probable typo: does ES  actually mean WES ?

Author Response

The authors are grateful for the contributions of the Reviewer. The requested adjustment on page 8 has been done. Furthermore, a new version of the figure is available on page 4 of the manuscript. The highlighting in green was removed because it did not have any purpose.

Reviewer 2 Report

This is a very well-written case report, raising awareness on an important topic in medical genetics. Although the subject per se is not novel, the specific association of the two overlapping conditions identified in the proband are new, and potentially interesting to the readers. The mini-review on other cases in the literature is minimalistic, but very useful as a collection of references.

Only some minor spell check is required - just a few mistakes and typos, for example:

Section 3.1, lines 2-3: "of healthy consanguineous parents" instead of "of health and consanguineous parents" Section 3.2, line 2: "added up to" or "totaled" instead of just "summed" Section 3.2, line 8: too many parentheses Section 3.2, fourth line from the bottom: "(76 of 105)" or "(76 out of 105)" instead of "(76 from 105)" Section 4: all gene names should be written in Italics, as was correctly done in the other sections.  

Author Response

The authors are grateful for the contributions of the Reviewer. All suggested minor adjustments were done and are marked up using the “Track Changes” function of the MS Word document.

Reviewer 3 Report

Paper for review

Correia-Costa et al. report on Dual molecular diagnoses of recessive disorders in a child from consanguineous parents

They reported variants in two different genes in the same individual.

The clinical and molecular data very are very interesting and could be of interest to the readers.

General comments:

#Abstract is missing? Why? Add abstract.

+ Follow nomenclature: https://varnomen.hgvs.org/

*- The reference sequence (NP_) and NM_ should also be added.

**Gene name should be italics. Kindly check.

IRB approval number should be mentioned.

Introduction:

Divide it into paragraphs, making it easy for the readers to understand.

Materials and Methods

Add headings…

Add a flow sheet about the WES filtration steps, was any other variant identified?

Results

Add subheadings like CMA, WES, Sanger, etc

How the variants were classified as pathogenic ACMG criteria fulfilled?

Cover the patient’s eyes with a black strip.

Was consent obtained about patient pictures publishing?

Discussion

Were any medications given?

How can we prevent the disorder?

A paragraph on future perspectives should be added. For example:

·         Proper genetic counseling, the introduction of the newborn screening program, and parenteral diagnosis can significantly reduce the burden of such severe skin disorders. This can be accomplished by prenatal genetic testing for monogenetic disorders (PGT-M). PGT and in vitro fertilization are options for parents wishing to have future pregnancies (PMID: 36406136). Although there is no specific management for these cases, patients are treated with supportive treatment.

 References:

·         Kindly cite: PMID: 29321360, DOI: 10.1007/s12041-017-0868-6

Author Response

We are thankful for your insightful considerations and certain that they have significantly improved de quality of our manuscript. In the enclosed file are our point-by-point responses to your comments.
